# Characterization of a Novel Rice Dynamic Narrow-Rolled Leaf Mutant with Deficiencies in Aromatic Amino Acids

**DOI:** 10.3390/ijms21041521

**Published:** 2020-02-23

**Authors:** Huimei Wang, Yongfeng Shi, Xiaobo Zhang, Xia Xu, Jian-Li Wu

**Affiliations:** State Key Laboratory of Rice Biology, China National Rice Research Institute, Hangzhou 310006, China; wangyingkai2006@126.com (H.W.); shiyongfeng@caas.cn (Y.S.); zhangxiaobo@caas.cn (X.Z.); mailxuxia@163.com (X.X.)

**Keywords:** narrow-rolled leaf, DAHPS, map-based cloning, rice (*Oryza sativa* L.)

## Abstract

The leaf blade is the main photosynthetic organ and its morphology is related to light energy capture and conversion efficiency. We isolated a novel rice *Dynamic Narrow-Rolled Leaf 1* (*dnrl1*) mutant showing reduced width of leaf blades, rolled leaves and lower chlorophyll content. The narrow-rolled leaf phenotype resulted from the reduced number of small longitudinal veins per leaf, smaller size and irregular arrangement of bulliform cells compared with the wild-type. *DNRL1* was mapped to chromosome 7 and encoded a putative 3-deoxy-7-phosphoheptulonate synthase (DAHPS) which catalyzes the conversion of phosphoenolpyruvate and D-erythrose 4-phosphate to DAHP and phosphate. Sequence analysis revealed that a single base substitution (T–A) was detected in *dnrl1*, leading to a single amino acid change (L376H) in the coding protein. The mutation led to a lower expression level of *DNRL1* as well as the lower activity of DAHPS in the mutant compared with the wild type. Genetic complementation and over-expression of *DNRL1* could rescue the narrow-rolled phenotype. *DNRL1* was constitutively expressed in all tested organs and exhibited different expression patterns from other narrow-rolled leaf genes. DNRL1-GFP located to chloroplasts. The lower level of chlorophyll in *dnrl1* was associated with the downregulation of the genes responsible for chlorophyll biosynthesis and photosynthesis. Furthermore, *dnrl1* showed significantly reduced levels of aromatic amino acids including Trp, Phe and Tyr. We conclude that OsDAHPS, encoded by *DNRL1*, plays a critical role in leaf morphogenesis by mediating the biosynthesis of amino acids in rice.

## 1. Introduction

Rice leaf morphology is critical for light energy capture and conversion efficiency. As a densely planted crop, a moderate leaf shape is conducive to maintaining its upright gesture, improving photosynthetic efficiency and increasing rice grain production [1,2]. A narrow leaf blade helps to maintain an upright gesture and has been proposed as one of the ideal characteristics in rice breeding practice [3]. Besides the natural germplasm, many genetically stable narrow leaf mutants have been obtained through physical, chemical and biological mutagenesis. So far, more than 30 genes mediating narrow leaf occurrence have been mapped and 17 of them have been cloned [4].

Narrow leaf mutants are usually accompanied by other phenotypic variations such as rolled leaf, dwarfism, increased tiller number and leaf angle. Based on the information from cloned narrow leaf genes, the molecular mechanisms underlying narrow leaf phenotypes are complicated and mainly associated with microRNAs, transcriptional factors, cellulose enzymes and phytohormones. *OsDCL1* encodes a dicer or dicer-like (DCL) protein which is required for the maturation of miRNAs and siRNAs. It has been shown that the weak loss of function *OsDCL1* transformants display narrow, rolled, and outward-folded leaves. In fact, *OsDCL1* plays not only a critical role in miRNA processing, but also functions in the immunity to rice blast fungi [5,6]. *NAL2* and *NAL3* are paralogs that encode an identical *OsWOX3A* transcriptional activator, which is involved in rice organ development covering lateral axis outgrowth and vascular patterning in leaves, lemma and palea morphogenesis in spikelets, and development of tillers and lateral roots [7]. *SLL1* encodes a SHAQKYF class MYB family transcriptional factor belonging to the KANADI family. The *sll1* mutant exhibits extremely incurved leaves due to the defective development of sclerenchymatous cells on the abaxial side [8].

*NRL1* [9,10], also known as *ND1* [11], *sle1* [12], *DNL1* [13] and *OsCD1* [14], encodes the cellulose synthase-like protein D4 (OsCslD4). Compared with the wild type, the *nrl1* mutant is characterized by narrow leaves, dwarfism, and a lower number of longitudinal veins. The *nal9* mutant shows not only narrow leaves, but also light green seedling leaves, reduced plant height, increased number of tillers and small size of panicles [15]. *NAL9* encodes an ATP-dependent Clp protease proteolytic subunit [16]. *OsCOW1* (*Constitutively wilted 1*), a member of the rice YUCCA gene family, is required for maintaining water homeostasis and an appropriate root/shoot ratio. The mutation of *OsCOW1* exhibits rolled leaves and reduced leaf width [17].

Similar to *nrl1*, *nal1* also contains a lower number of longitudinal veins. *NAL1* is richly expressed in vascular tissues and modulates lateral leaf growth by affecting polar auxin transport as well as the vascular patterns of rice plants [18]. Furthermore, it also has been suggested that *NAL1* controls leaf width and plant height by affecting cell division and expansion [19,20]. The *nal7* mutant shows a milder effect on leaf width and vein number, and both the large and small veins are similarly affected [21]. *NAL7* encodes a favin-containing monooxygenase and is likely involved in auxin biosynthesis as the IAA content in *nal7* mutant is altered compared with the wild type. The *nal7* mutant overexpressing *NAL7* cDNA exhibits overgrowth and abnormal root morphology, which is likely caused by auxin overproduction [22]. The *tdd1* plants exhibit pleiotropic phenotypes such as dwarfism, narrow leaves, shorter roots and abnormal flowers. *TDD1* encodes a protein homologous to the anthranilate synthase b-subunit, which catalyzes the first step of the Trp biosynthesis pathway and functions upstream of Trp-dependent IAA biosynthesis [23]. The *shallot-like 2* rolling leaf mutant (*sll2*) exhibits adaxially rolled leaves accompanied by increased photosynthesis and reduced plant height and tiller number. The shrinkage of bulliform cells is considered responsible for the formation of inward-curved leaves [24]. *NRL2*, also known as *SRL2* and *AVB*, encodes a novel protein with unknown biochemical function [25]. SRL2 plays an important role in regulating leaf development, particularly during sclerenchymatous cell differentiation [26]. AVB is involved in the maintenance of the normal cell division pattern in lateral primordia development and procambium establishment associated with auxin signaling [27].

Here we report a novel *dynamic narrow-rolled leaf 1* (*dnrl1*) mutant obtained from an EMS-induced rice mutant library. We conducted agronomic trait evaluation, physiological and biochemical analysis, genetic analysis, gene mapping and cloning, functional complementation, subcellular location and gene expression analysis. We conclude that OsDAHPS, encoded by *DNRL1,* plays a critical role in leaf morphogenesis by modulating the biosynthesis of amino acids. 

## 2. Results

### 2.1. Phenotype of dnrl1 Mutant

A *dynamic narrow-rolled leaf* mutant, *dnrl1*, was derived from an EMS-induced IR64 (The wild-type, WT) mutant bank. Under natural field conditions, the narrow and rolled leaf phenotype of *dnrl1* was observed 10 days after sowing (DAS) and reached the maximum rolling index (>90%) at approximately 30 DAS (Figure 1A–H). The narrow leaf phenotype of *dnrl1* lasted throughout the whole growth period (Figure 1A) while the rolled leaf phenotype was gradually recovered after 30 DAS (Figure 1B) and eventually became flat at 100 DAS at the heading stage (Figure 1B,H). During the whole growth duration, *dnrl1* also showed a slight green leaf phenotype. We thus measured the chlorophyll contents of *dnrl1* and WT at the seedling stage (20 DAS). The results showed that chlorophyll a, chlorophyll b and total chlorophyll contents in *dnrl1* were reduced by about 8.8%, 12.4% and 9.8%, respectively, compared with WT (Figure 1I). In addition, *dnrl1* exhibited a fewer number of small veins (intermediate veins) compared with WT; however, the number of large veins was similar between *dnrl1* and WT (Figure 1J,K).

### 2.2. Genetic Control and Fine Mapping of DNRL1

To determine the genetic control of the dynamic narrow-rolled trait, F_1_ plants and F_2_ populations were analyzed. The results showed that all F_1_ plants derived from the two crosses (*dnrl1*/IR64 and *dnrl1*/Moroberekan) showed flat leaves, indicating that the mutation was controlled by a recessive gene(s). To determine the number of genes, F_2_ individuals derived from the two crosses *dnrl1*/IR64 and *dnrl1*/Moroberekan were further phenotyped. Among the 833 F_2_ individuals derived from *dnrl1*/IR64, 632 plants were flat-leaved and 201 plants showed dynamic narrow-rolled leaves, fitting to a 3:1 ratio (χ2 = 0.34 < χ2_0.05_ = 3.84). Among the 2353 plants derived from *dnrl1*/Moroberekan, 1752 plants were normal-leaved and 601 plants were dynamic narrow-roll leaved, again fitting to the expected 3:1 ratio (χ2 = 0.37 < χ2_0.05_ = 3.84). The results indicated that the mutation was controlled by a single recessive gene (Table 1).

To primarily map the mutation, bulked segregate analysis was adopted for parental and DNA pool screening using 240 SSR markers randomly distributed on the rice genome and 36 F_2_ plants derived from *dnrl1*/Moroberekan were firstly used for the confirmation of a potential linkage. The results showed that *DNRL1* was initially located between RM234 and RM134 on chromosome 7. To further confirm and delimit the mutation, 144 randomly chosen F_2_ dynamic narrow-rolled leaf plants were genotyped using RM234 and RM134, and the result showed that the number of crossing-over plants to the corresponding markers were 10 and 37, respectively. Then, three markers (RM18, RM8261 and RM5688) distributed within the region were selected for further genotyping. The results showed that the number of crossing-over plants to the corresponding three markers was 4, 9, and 26, respectively. Thus, *DNRL1* was delimited to the region flanked by RM18 and RM8261. To fine map the candidate gene, six markers including three polymorphic SSR markers (RM21981, RM21985 and RM21989) and 3 InDel markers (S1, S5 and S11) were then used for genotyping a total of 601 dynamic narrow-rolled F_2_ individuals derived from *dnrl1*/Moroberekan (Appendix A). As a result, *DNRL1* was finally delimited to a region covering a physical distance of 43.2 kb between S1 and RM21981 (Figure 2A).

Based on the gene annotation in the NCBI and Gramene database, four open reading frames (ORFs) (LOC_Os07g42924, LOC_Os07g42940, LOC_Os07g42950 and LOC_Os07g42960) are predicted in the 43.2 kb region (Figure 2B). Sequence analysis of the ORFs showed that only one nonsynonymous mutation, T1127A (L376H), was detected in LOC_Os07g42960 (Figure 2C). No mutations were found in the other three ORFs. The 3591 bp LOC_Os07g42960 contains five exons and four introns and is considered as the most likely candidate gene responsible for the dynamic narrow-rolled leaf phenotype of *dnrl1*. Interestingly, LOC_Os07g42960 has another shorter transcript, LOC_Os07g42960.2, harboring a nonsynonymous mutation at position T584A (L195H) (Figure 2C).

### 2.3. DNRL1 Complements the dnrl1 Phenotype

To validate the candidate gene, the complementary vector pCAMBIA1300-dnrlC was introduced into the embryogenic calli derived from *dnrl1* mature seeds via *Agrobacterium tumefaciens*-mediated transformation. A total of 13 positive transformants were obtained and 11 of them exhibited a normal flat leaf phenotype throughout the whole growth period. The results indicated that LOC_Os07g42960 could rescue the dynamic narrow-rolled phenotype and was indeed the target gene (Figure 3A–D,G–J,W).

As mentioned above, LOC_Os07g42960 consists of two transcripts, LOC_Os07g42960 and LOC_Os07g42960.2 (Figure 2C). The overexpression vectors pu1301-dnrlox1 and pu1301-dnrlox2 were introduced into the embryogenic calli of *dnrl1* via *Agrobacterium tumefaciens*-mediated transformation. For pu1301-dnrlox1 transformation, a total of seven positive transformants were obtained and all exhibited the normal flat leaf phenotype throughout the whole growth period (Figure 3E,F,K,L). In contrast, all of the five pu1301-dnrlox2 positive transformants showed narrow-rolled leaves (data not show). The complementation and over-expression transformants showed a normal flat leaf phenotype similar to WT while some agronomic traits such as plant height and 1000-grain weight were not fully recovered compared with WT (Appendix A). Nevertheless, our results demonstrated that both genomic LOC_Os07g42960 and the correspondence CDS of LOC_Os07g42960 could rescue the narrow-rolled leaf phenotype. In contrast, LOC_Os07g42960.2 could not rescue the phenotype and the function of LOC_Os07g42960.2 has yet to be investigated.

To characterize and compare *dnrl1* (transgenic plants and WT), we firstly observed the leaf microstructure. We found that the size of bulliform cells was smaller in *dnrl1* than that of WT (Figure 3M,N). The bulliform cells in genetic complementation and over expression lines were normal in size and closely arranged similar to those of WT (Figure 3O–R). In addition, the number of large veins and small veins in complementation lines and overexpression lines were all recovered to the WT level (Figure 3S–U). The decreased number of small veins in *dnrl1* was likely due to the decreased number of small veins between two adjacent large veins (Figure 3U). We lastly checked whether the chlorophyll level could be recovered in the complementation lines. The results showed that chlorophyll a, chlorophyll b and total chlorophyll contents in the two complementation lines were recovered to the similar level of WT in nine-week-old plants (Figure 3V). Taken together, *DNRL1* was the target gene responsible for the narrow-rolled leaf phenotype that directly resulted from a decreased number of small leaf veins and the abnormal shape of bulliform cells.

### 2.4. DNRL1 Encodes a DAHP Synth II Enzyme and Modulates Aromatic Amino Acid Synthesis

The NCBI conservative structure domain analysis (CDSEARCH/CDD v3.16) [28] shows that *DNRL1* encodes a 3-deoxy-7-phosphoheptulonate synthase (DAHPS) which catalyzes the conversion of phosphoenolpyruvate and D-erythrose 4-phosphate to DAHP and phosphate. DNRL1 belongs to the DAHP synth II superfamily. There are two *DAHPS* in rice, *OsDAHPS1* and *OsDAHPS2,* which locate to chromosome 7 and 3, respectively. The nucleotide sequences of *OsDAHPS1* and *OsDAHPS2* are highly homologous (83%), and the amino acid sequences of *OsDAHPS1* and *OsDAHPS2* are highly conserved in plant DAHPS identified in *Arabidopsis*, potato and tomato [29]. DAHPS is the first key enzyme of the shikimate pathway (SP) for the biosynthesis of aromatic amino acids and their secondary metabolites in plants.

To determine whether the mutation affects the enzymatic activity of DAHPS, we evaluated DAHPS activity in IR64, *dnrl1*, complementation and over-expression plants. The results showed that the activity of DAHPS was significantly decreased in *dnrl1* compared with IR64, while the DAHPS activities in complementation and overexpression lines were recovered to a similar level to the WT (Figure 4A). To determine whether the mutation affects the synthesis of aromatic amino acids, we carried out free amino acid content measurement. The results indicated that the levels of phenylalanine, tyrosine and tryptophan in *dnrl1* were significantly decreased compared with WT, respectively (Figure 4B–D). Furthermore, the contents of glycine and lysine in *dnrl1* were similar to WT, while the level of aspartic acid increased in *dnrl1* compared to WT (Figure 4E–G). The results demonstrated that the mutation greatly impeded the activity of DAHPS and the normal biosynthesis of aromatic amino acids.

### 2.5. DNRL1 Expresses Constitutively

To clarify the expression pattern of *DNRL1*, quantitative reverse transcription polymerase chain reaction (qRT-PCR) analysis was performed using the total RNA derived from the roots, stems, internodes, leaves, leaf sheaths and panicles from IR64. The results showed that *DNRL1* was expressed in all the organs tested with the highest expression level in shoots at the seedling stage, in roots at the tillering stage, and in panicles at the filling stage (Figure 5). The results showed that *DNRL1* was constitutively expressed in rice.

### 2.6. DNRL1-GFP Localizes to Chloroplasts

Following the determination of the expression pattern of *DNRL1*, we carried out subcellular location analysis of DNRL1-GFP in rice cells. Using the protoplast transient transformation of rice, pAN580 and pOsDnrl1-GFP were introduced into the protoplasts, respectively. As shown in Figure 6, the fluorescence signal distribution of DNRL1-GFP is co-localized with chloroplast spontaneous red fluorescence. The results demonstrated that DNRL1 mainly located to chloroplasts.

### 2.7. Altered Expression of Leaf Morphology and Chlorophyll Biosynthesis-Associated Genes in dnrl1

To investigate whether the expression of leaf morphology-associated genes were affected in the mutant, we measured the transcript levels of *NAL1*, *NRL1*, *OsCOW1* and *TDD1* by qRT-PCR. The results showed that the transcript level of *DNRL1* itself was significantly decreased in *dnrl1* and recovered to the WT level in complementation lines (Figure 7A). The transcript levels of two narrow leaf genes, *NAL1* and *TDD1,* were also significantly decreased in *dnrl1*, as well as in the complementation lines (Figure 7B,E). In contrast to *NAL1* and *TDD1*, the transcript levels of *NRL1* and *OsCOW1* were significantly increased in *dnrl1* and were unable to be recovered to the WT levels in the complementation lines (Figure 7C,D). These results suggested that the DNRL1-regulated narrow-leaf mechanism was likely different from the NAL1, TDD1, NRL1, and OsCOW1 regulated mechanisms.

As mentioned above, *dnrl1* was chlorophyll-deficient and showed a lower seed setting rate and 1000-grain weight compared with WT (Figure 1I, Appendix A). We thus checked the expression level of genes associated with chlorophyll biosynthesis and photosynthesis in the mutant through qRT-PCR analysis. The results showed that 9 out of the 10 investigated chlorophyll synthetic genes (*OsCAO1* and *OsCAO2* encode chlorophyll a oxygenase; *OsPORA* and *OsPORB* encode NADPH-dependent protochlorophyllide oxidoreductase; *LYL1* encodes geranylgeranyl reductase; *OsChlD* encodes the magnesium-chelatase D subunit; *YGL1* encodes chlorophyll synthase; *YGL8* encodes the catalytic subunit of magnesium-protoporphyrin IX monomethyl ester cyclase; and *HEMA1* encodes glutamyl tRNA reductase) were downregulated in *dnrl1* compared with WT (Figure 7F). We also carried out chloroplast ultrastructure observation on IR64, *dnrl1*, dnrl1-c1 and dnrl1-ox3, and found that there was no significant change in chloroplast morphology except that *dnrl1* had a fewer number of chloroplasts compared with IR64 and dnrl1-c1 (Appendix A). In addition, all seven photosynthesis or chloroplast biogenesis-related genes (*OscpSRP54* encoding for the chloroplast signal recognition particle 54 kDa subunit; *SPP* encoding for the rice stromal processing peptidase; *OsRpoTp* encoding for plastid RNA polymerase; *rpoA* encoding for the plastid RNA polymerase alpha chain; *PPR1* encoding for the chloroplast pentatricopeptide repeat protein; *psbA* encoding for photosystem II protein D1; and *psaA* encoding for photosystem I P700 chlorophyll A apoprotein A1) were significantly downregulated in *dnrl1* compared with WT (Appendix A). Our results clearly demonstrated that the *DNRL1* mutation caused a huge variation in the expression of genes associated with chlorophyll biosynthesis and chloroplast development leading to the narrow-rolled leaf phenotype in *dnrl1*.

## 3. Discussion

Narrow leaf mutants are a useful source for the study of mechanisms underlying leaf development in plants. So far, more than 30 genes related to the narrow leaf phenotype have been mapped and 17 of them have been cloned in rice [4]. A narrow leaf blade is usually associated with an abnormal number of longitudinal leaf veins. The *nal1* mutant exhibits a reduced number of vertical leaf veins with an abnormal arrangement of vascular bundles [18]. Unlike *nal1*, *nrl1* shows not only narrowed but also semi-rolled leaves. The number of longitudinal veins in *nrl1* is decreased, with smaller sized abaxial bulliform cells [9]. Unlike *nal1* and *nrl1*, the narrow and rolled leaf phenotype of *Oscow1* is caused by cellular water evaporation rather than structural and cellular defects [17]. The *tdd1* mutant displays multiple morphological defects including the narrow leaf trait, which can be recovered by exogenous treatment with tryptophan and IAA [23]. In the present study, we identified a dynamic narrow rolled-leaf mutant, *dnrl1*. Compared to the wild type, the *dnrl1* mutant had 25% fewer small veins (intermediate veins), although the number of larger veins was similar between the two genotypes. Therefore, we conclude that DNRL1 modulates the number of small veins in rice leaf blades. Furthermore, the bulliform cells were smaller in size and arranged irregularly in *dnrl1*, indicating that DNRL1 also controls the size and distribution of bulliform cells in rice. It was noticed that *dnrl1* was deficient in chlorophyll content due to the downregulation of a number of chlorophyll biosynthesis genes which, in contrast, are upregulated in *nrl3* [30]. Interestingly, only *YL1* was upregulated while the other genes tested were downregulated in *dnrl1*. Obviously, the upregulation of *YL1* did not contribute to the light pale-green phenotype of *dnrl1*. Nevertheless, the relationship between the chlorophyll level and leaf shape is largely elusive and required further studied both in *dnrl1* and *nrl3*. Notably, the *dnrl1* phenotype changed dynamically at different developmental stages. This dynamically changed phenotype is unique to *dnrl1* and has not been reported in other rice narrow-rolled leaf mutants. Thus, we hypothesize that *DNRL1* may have a different molecular mechanism modulating leaf development.

Based on the information from cloned narrow-rolled leaf genes, the molecular mechanisms underlying narrow-rolled leaf phenotypes are complicated. It is generally considered that the curling of rice leaves is mainly controlled by the leaf curling genes responsible for the development of adaxial/abaxial cells, vesicular cells, cell expansion and osmotic pressure [31]. On the other hand, the narrow leaf genes mainly affect the number and width of vascular bundles of leaves by regulating auxin synthesis, polar transport and distribution [10,18,22]. *NAL1* is highly expressed in vascular tissues and modulates lateral leaf growth by affecting polar auxin transport as well as the vascular patterns in rice plants [18]. *NAL7* encodes a favin-containing monooxygenase and is likely involved in auxin biosynthesis as the IAA content in *nal7* is altered compared with the wild type. In addition, *nal7* overexpressing *NAL7* cDNA exhibits overgrowth and abnormal root morphology, which is likely caused by auxin overproduction [22]. *TDD1* encodes a protein homologous to the anthranilate synthase b-subunit, which catalyzes the first step of the tryptophan biosynthesis pathway and functions upstream of tryptophan-dependent IAA biosynthesis [23]. *OsCOW1* (*Constitutively wilted 1*) is a member of the rice YUCCA gene family which is required for maintaining water homeostasis and an appropriate root/shoot ratio. Unlike most narrow/rolled leaf genes mentioned above, *OsDNRL1* was not only responsible for the narrow leaf phenotype, but also the rolled leaf phenotype. *OsDNRL1* encodes a 3-deoxy-7-phosphoheptulonate synthase (DAHPS), which is the first key enzyme of the shikimate pathway. DAHPS catalyzes the conversion of phosphoenolpyruvate and D-erythrose 4-phosphate (E4P) to 3-deoxy-D-arabinoheptulosonate 7-phosphate (DAHP) and phosphate. The shikimate pathway provides carbon skeletons for the synthesis of aromatic amino acids l-tryptophan, l-phenylalanine, and l-tyrosine. It is a high flux bearing pathway and it has been estimated that between 20–50% of fixed carbon is directed through the pathway [32,33,34]. The mutation in *dnrl1* resulted in both decreased expression levels of *DNRL1* and lowered enzymatic activity of DAHPS, although the mechanism has yet to be clarified. Furthermore, *dnrl1* was deficient in Trp, Phe and Tyr. Therefore, we conclude that OsDAHPS encoded by *DNRL1* plays a critical role in leaf morphogenesis by mediating the biosynthesis of aromatic amino acids in rice.

Tryptophan is the precursor of IAA biosynthesis, and thus the level of IAA could theoretically be affected in the tryptophan-deficient *dnrl1*. Indeed, we treated the three-week-old mutants with exogenous IAA and found that the IAA treatment with different gradients could not restore the narrow-rolled leaf phenotype (data not shown), indicating the application of exogenous IAA could not rescue the narrow-rolled leaf phenotype in *dnrl1.* This is quite different from *tdd1,* whose phenotype is able to be recovered by exogenous IAA and Trp treatments [23]. Unfortunately, we were not able to measure the endogenous content of IAA in *dnrl1* because of the repeatability of IAA values. Therefore, whether IAA is responsible for the narrow-rolled phenotype in *dnrl1* has yet to be clarified. Nevertheless, the expression of leaf morphology-associated genes including *NAL1*, *NRL1*, *OsCOW1* and *TDD1* in the wild-type, *dnrl1*, complementation and overexpression lines demonstrated that the DNRL1-regulated dynamic narrow-rolled leaf mechanism was likely different from those of *NAL1*, *TDD1*, *NRL1*, *OsCOW1,* as well as *OsWOX4* and *SNFL1*-meadiated mechanisms [35,36]. These differences in regulating leaf development are further supported by the fact that DNRL1 localized to chloroplasts while none of NAL1 (nucleus and cytoplasm), NRL1 (Golgi body), OsCOW1 (probably ER), OsWOX4 (nucleus) and SNFL1 (nucleus) locate to chloroplasts, except TDD1 for which the subcellular location has yet to be confirmed [16,17,22,23,35,36]. Thus, the isolation of *dnrl1* and preliminary functional characterization of DNRL1 might provide a novel source and insight for characterization of leaf development in rice.

## 4. Materials and Methods

### 4.1. Plant Material

The dynamic narrow-rolled leaf mutant *dnrl1* (originally named HM28) was obtained from ethyl methane sulfonate (EMS) mutagenesis of the indica cultivar IR64 mature seed [37]. The mutant has been selfed for more than nine generations and the dynamic narrow-rolled leaf phenotype has been stably inherited in greenhouse and field conditions in Hangzhou, Zhejiang and in Lingshui, Hainan, China. The normal flat leaf cultivar Moroberekan was used as the male parent to cross with the mutant for the construction of F_2_ populations for genetic analysis and gene mapping. The parents and the population were grown in the paddy field at the China National Rice Research Institute (CNRRI) in the regular season. The transgenic and wild type plants were grown in the net house at the CNRRI.

### 4.2. Map-Based Cloning of DNRL1

For genetic analysis, we crossed *dnrl1* with IR64 and Moroberekan in the Hangzhou Experimental Station at CNRRI in 2014. F_1_ plants were grown at the Lingshui Experimental Station in 2015 and selfed to generate F_2_ populations. F_2_ individuals from the two crosses were grown in Hangzhou in the same year and phenotyped for segregation analysis.

The F_2_ population derived from the *dnrl1*/Moroberekan cross was chosen for mapping of the mutation. Bulked segregant analysis was first used to rapidly locate the mutation on a chromosome and a physical linkage map was then constructed using additional markers surrounding the mutation. An equal amount of leaf blades from 10 wild type plants and 10 mutant type plants were collected for DNA extraction to form a wild type DNA pool and a mutant DNA pool, respectively. DNA of the parents and F_2_ individuals with the mutant phenotype were extracted following the mini-preparation method [38]. Simple sequence repeat (SSR) markers were obtained from (http://www.gramene.org/) while insertion/deletion (InDel) markers were designed using Primer 5.0 after comparison of the sequences between the japonica cultivar Nipponbare and the indica cultivar 9311 in the following public databases: RGP (http://rgp.dna.affrc.go.jp/E/toppage.html), Gramene (http://gramene.org/genome_browser/index.html) and the Gene Research Center of the Chinese Academy of Sciences (http://rice.genomics.org.cn/rice/index2.jsp). The primers were synthesized by Tsingke Biotech Co. Ltd (Hangzhou, China) and listed in Appendix A. PCR reactions and detection were carried out as described previously [39].

### 4.3. Genetic Complementation and Over-Expression Analysis

For complementation of the mutant phenotype, a 5.5 kb wild type genomic fragment, containing the entire 3.2 kb open reading frame (ORF) of *dnrl1*, a 1.1 kb upstream region, and a 1.2 kb downstream region was amplified by PCR using the specific primers DNRLF/R (DNRLF, TCCCCCGGGGGATAGATGTTGACAAGTGGC and DNRLR, GCTCTAGA TTCATTTGAGTTGCTGGGTT). The PCR product was double digested with *Sma I* and *Xba I*, and the fragment was recovered using an Axygen DNA Gel Extraction Kit (Axygen Scientific, San Francisco, CA, USA). Then, the fragment was cloned into the binary vector pCAMBIA1300 to form a new transformation construct, pCAMBIA1300-dnrlC. The construct was introduced into the embryogenic calli derived from the mature seed of *dnrl1* using *Agrobacterium*-mediated transformation [40].

For over-expression analysis, the 1.6 kb CDS of LOC_Os07g42960 and 1.1 kb CDS of LOC_Os07g42960.2 were amplified by PCR using the specific primers DNRLox1F/R (DNRLox1F, ATGGGTACCAATGGCGCTCGCCACCAACT/R, DNRLox1R, ATGGGATTCAGCAGTTTAGAAAGCCAATG) and DNRLox2F/R (DNRLox2F, ATGGGTACCAATGGCTGGCCAGTTCGCCAA, DNRLox2R, ATGGGATTCAGCAGTTTAGAAAGCCAATG), respectively. The PCR products were double digested with *Kpn I* and *BamH I*, and the fragments were recovered using an Axygen DNA Gel Extraction Kit (Axygen Scientific, San Francisco, CA, USA). Then, the fragments were cloned into the plant expression vector pu1301 to form two new transformation constructs, pu1301-dnrlox1 and pu1301-dnrlox2, respectively. The constructs were introduced into the embryogenic calli generated from the mature seed embryos of Nipponbare using *Agrobacterium*-mediated transformation [40].

### 4.4. Measurement of Chlorophyll Content

Chlorophyll contents from 20-day-old seedlings of *dnrl1* WT and 9-week-old *dnrl1*, WT and complementation lines were determined according to the method described by Arnon [41]. The means from three biological replicates were used for analysis by Student’s *t* test.

### 4.5. Quantitative Reverse Transcription PCR (qRT-PCR)

To determine the expression profile of *DNRL1* in different tissues, total RNA was extracted from IR64 at the seedling stage (2 weeks after sowing), tillering stage (10 weeks after sowing) and the grain filling stage using the TRIzol method following the manufacture’s instruction. For qRT-PCR analysis of genes associated with chlorophyll biosynthesis, chloroplast development and leaf morphology associated genes, total RNA was extracted from leaves of *dnrl1* and IR64 at 10 weeks after sowing. The first-strand cDNA was synthesized using the First Strand cDNA Synthesis Kit following the manufacturer’s protocol (TOYOBO Biotech, Osaka, Japan). qRT-PCR was performed on a Thermal Cycle Dice TM Real Time System II (Takara Biotech, Japan). A ubiquitin gene was used as an internal control. The relative expression level of the tested genes was normalized to ubiquitin and calculated by the 2^−∆∆CT^ method [42]. All assays were repeated three times and the means were used for analysis by Student’s *t* test. Primers used for qRT-PCR are listed in Appendix A. 

### 4.6. Subcellular Localization

For subcellular localization analysis of the DNRL1 protein, the cDNA sequence corresponding to the coding sequence (CDS) of *DNRL1* was amplified from IR64 using the specific primer DNRSUC (forward: ATGTCTAGAATGGCGCTCGCCACCAACTCC; reverse: ATGGGATTCGAAAGCCAATGGGGGCAATGGCA). The PCR product was double digested with *Xba I* and *BamH I*, and the fragment was inserted into the 5’-terminal of GFP driven by the CaMV 35S promoter in the transient expression vector PAN580 to form a new construct, PAN580-DNRL1. The construct was transformed into rice protoplasts according to the protocol described previously [43].

### 4.7. Determination of Amino Acid Content

Fresh leaf powder (100 mg) from 9-week-old seedlings of IR64 and *dnrl1* were used for amino acid extraction. The amino acid content was determined using an automatic amino acid analyzer model Sykam S-430D (Sykam, München, Germany) following the method described by Mei et al. [44] and the determination procedures described previously [45]. The means from three biological replicates were used for data analysis by Student’s *t* test.

### 4.8. Assay of DAHPS Activity

Leaves (500 mg) from 9-week-old seedlings of IR64, *dnrl1*, complementation and over-expression lines were collected for analysis. The sample preparation and DAHPS activity assay were carried out according to method described by Wang et al. [46]. Absorption values were measured using a SpectraMax i3x Multi-Mode Microplate Reader (Molecular Devices, Sunnyvale, CA, USA). The means from three biological replicates were used for data analysis by Student’s *t* test.

## 5. Conclusions

We isolated a novel rice *Dynamic Narrow-Rolled leaf 1* (*dnrl1*) mutant and cloned the causal gene *DNRL1*, which encodes a putative 3-deoxy-7-phosphoheptulonate synthase (DAHPS) that catalyzes the conversion of phosphoenolpyruvate and D-erythrose 4-phosphate to DAHP and phosphate. A single base substitution (T-A) was detected in *dnrl1*, leading to a single amino acid change (L376H) in the coding protein. DNRL1 modulates the number of small leaf veins and the size and distribution of bulliform cells. The mutation led to a lower level expression of *DNRL1*, lower activity of DAHPS and reduced levels of aromatic amino acids including Trp, Phe and Tyr in *dnrl1*. We conclude that OsDAHPS encoded by *DNRL1* plays a critical role in leaf morphogenesis by mediating the biosynthesis of amino acids in rice.

## Figures and Tables

**Figure 1 ijms-21-01521-f001:**
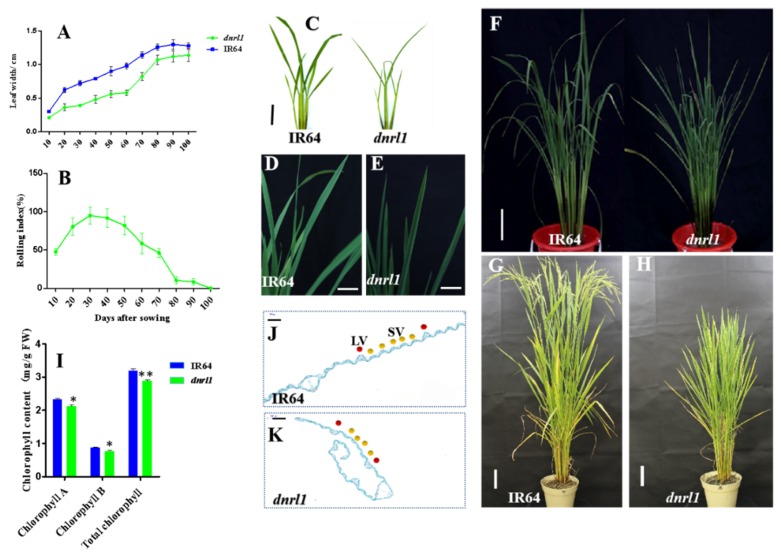
Performance of *dnrl1* under field conditions. (**A**) Leaf width of IR64 and *dnrl1,* the abscissa represents days after sowing; (**B**) dynamic leaf rolling index of *dnrl1*; (C-H) phenotype of IR64 and *dnrl1* at 4 weeks old ((**C**); bar = 2 cm) and 9 weeks old (**D**–**F**); (**D**) and (**E**), upper leaf blades of IR64 and *dnrl1* (for (**D**) and (**E**), bar = 2 cm; for (**F**), bar = 5 cm); 14-week-old plants of IR64 and *dnrl1* ((**G**) and (**H**), bar = 10 cm); (**I**) chlorophyll contents of IR64 and *dnrl1* at 20 days after sowing (DAS). Values are means ± standard deviation (SD) (*n* = 3), * indicates significance at *p* ≤ 0.05, ** indicates significance at *p* ≤ 0.01 by Student’s *t* test.; (**J**) and (**K**), cross-section of IR64 and *dnrl1* (for (**J**) and (**K**), bar = 200 μm), red dot indicates large vein (LV), yellow dot indicates small vein (SV).

**Figure 2 ijms-21-01521-f002:**
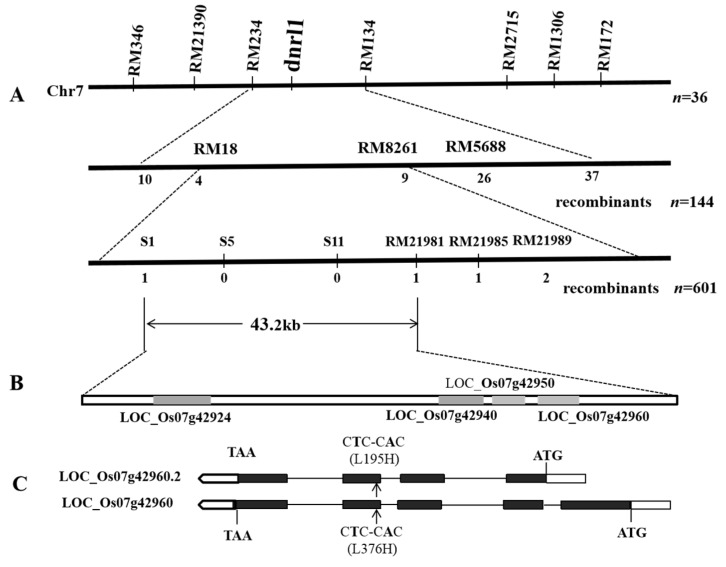
Map-based cloning of *DNRL1*. (**A**) The *DNRL1* locus was first mapped to chromosome 7 between markers RM234 and RM134, then narrowed down to an approximately 161 kb region between RM18 and RM8261 using 144 F_2_ mutant type individuals from the cross of *dnrl1*/Moroberekan. The *DNRL1* locus was further delimited to a 43.2 kb region between markers S1 and RM21981 using 601 F_2_ mutant type plants from the cross of dnrl1/Moroberekan; (**B**) a total of 4 ORFs (in gray box) were predicted in the region; (**C**) the candidate gene has two transcripts, LOC_Os07g42960 and LOC_Os07g42960.2 and a single nucleotide substitution from T to A at position 1127 of LOC_Os07g42960 and position 584 of LOC_Os07g42960.2. CDS were identified, respectively. Black boxes: exons; blank box: untranslated region; line: intron; arrow indicates the mutant site. Markers used for fine mapping are listed in Appendix A.

**Figure 3 ijms-21-01521-f003:**
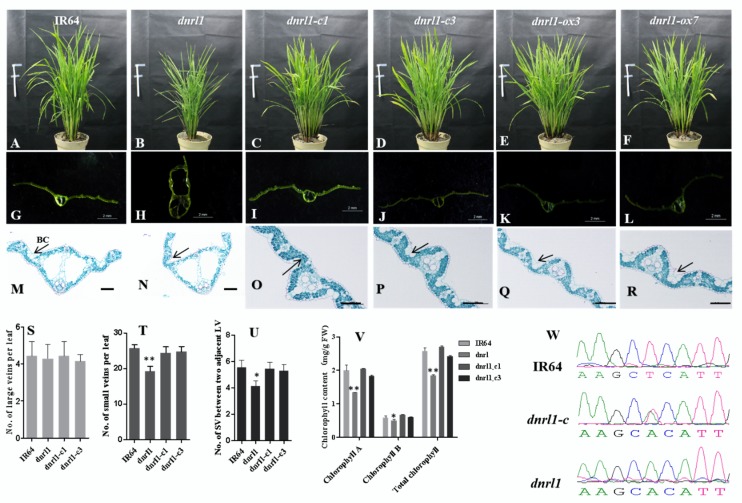
Functional complementation of the mutation. (**A**–**F**) Nine-week-old IR64 (WT), *dnrl1*, complemented and over-expression plants, bar = 10 cm; (**G**–**L**) leaf blade transverse section of (**A**–**F**), bar = 2 mm; (**M**–**R**) slice observation of (**G**–**L**) (black arrow indicates bulliform cells, BC), for (**M**) and (**N**), bar = 50 μm; for (**O**–**R**), bar = 100 μm); (**S**) number of LV per leaf; (**T**) number of SV per leaf; (**U**) number of SV between two adjacent LV; (**V**) chlorophyll contents of IR64, *dnrl1*, dnrl1-c1 and dnrl1-c3 at nine weeks. Values are means ±SD (*n* = 3); *indicates significance at *p* ≤ 0.05; ** indicates significance at *p* ≤ 0.01 by Student’s *t* test; (**W**) the nucleotide at the mutation site in IR64, *dnrl1*, and complemented plants.

**Figure 4 ijms-21-01521-f004:**
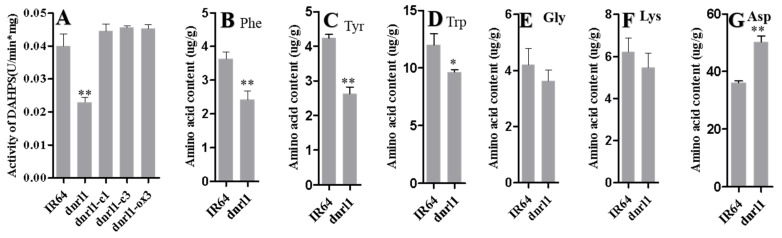
Comparison of 3-deoxy-7-phosphoheptulonate synthase (DAHPS) activity and amino acid contents. (**A**) DAHPS activity assay; (**B**–**G**) amino acid contents of IR64 and *dnrl1.* Nine-week-old seedling leaves were used for the DAHPS activity assay and amino acid content determination. IR64, wild type; *dnrl1*, mutant; dnrl1-c1 and dnrl1-c3, complementation lines; dnrl1-ox3, over-expression line. Values indicate means ±SD (*n* = 3); * indicates significance at *p* ≤ 0.05; ** indicates significance *p* ≤ 0.01 by Student’s *t* test.

**Figure 5 ijms-21-01521-f005:**
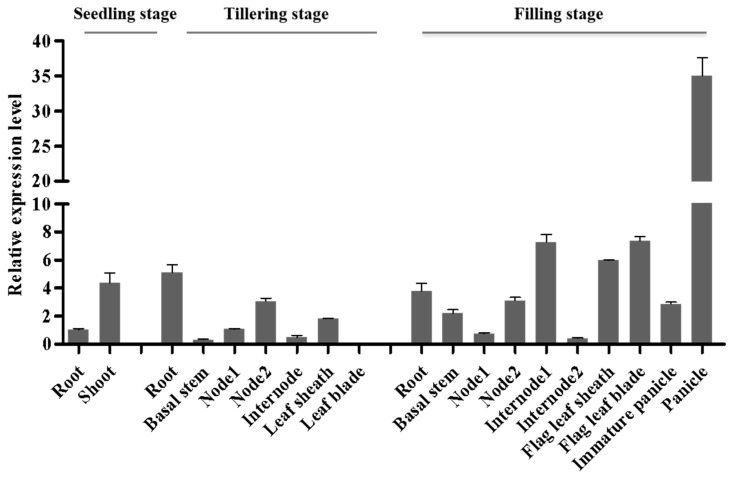
Relative expression levels of *DNRL1* in various rice tissues at different growth stages. qRT-PCR analysis of *DNRL1* in the wild-type.

**Figure 6 ijms-21-01521-f006:**
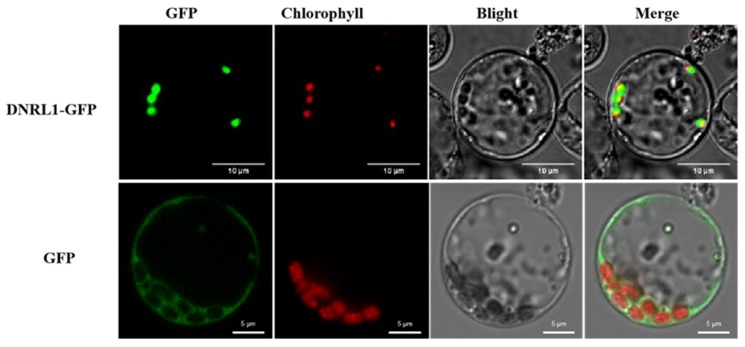
Subcellular location of DNRL1-GFP at chloroplasts. Subcellular location of DNRL1-GFP was performed using rice protoplasts. GFP, green fluorescence protein. For DNRL1-GFP, bar = 10 μm; for GFP, bar = 5 μm.

**Figure 7 ijms-21-01521-f007:**
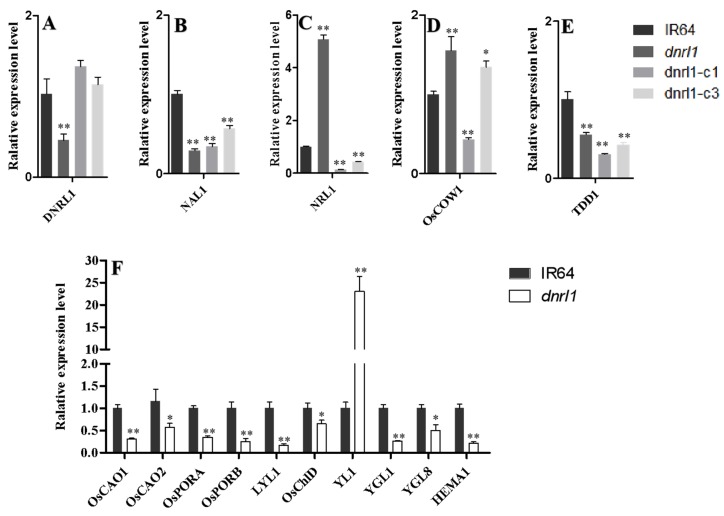
qRT-PCR analysis of leaf morphology and chlorophyll biosynthesis-associated genes. (**A**–**E**) Leaf morphology-associated genes; (**F**) chlorophyll biosynthesis-associated genes. The accession numbers and primer sequences of these genes are listed in Appendix A. Values indicate means ±SD (*n* = 3); * indicates significance at *p* ≤ 0.05; ** indicates significance at *p* ≤ 0.01 by Student’s *t* test.

**Table 1 ijms-21-01521-t001:** Genetic control of the narrow-rolled leaf trait in *dnrl1*.

Cross	F_1_ Phenotype	F_2_			*P* _(3:1)_	χ2 _0.05_
		Total No. of Plants	No. of Flat Leaf Plants	No. of Narrow Rolled-Leaf Plants		
*dnrll*/IR64	Flat leaf	833	632	201	0.56	0.34
*dnrll*/Moroberekan	Flat leaf	2353	1752	601	0.54	0.37

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
