# Peer review of "Characterization of a Novel Rice Dynamic Narrow-Rolled Leaf Mutant with Deficiencies in Aromatic Amino Acids"

_ijms, 2020, doi:10.3390/ijms21041521_

Round 1

Reviewer 1 Report

The manuscript entitled “Characterization of a novel rice dynamic narrow-rolled leaf mutant
with deficient in aromatic amino acids” by Huimei Wang described and reported  a narrow-rolled leaf 1 (dnrl1) mutant obtained from EMS-induced rice mutant library. After trait analysis, genetic analysis the authors concluded that OsDAHPS encoded by DNRL1 plays a role in leaf morphogenesis by modulating the biosynthesis of amino acids. The authors claimed that dnrl1 is novel and important.

dnrl1 mutants accounted for reduced width of leaf blades, rolled leaves and lower chlorophyll content. This gene has monogenic inheritance as reported. The authors mapped this gene in chromosome 7 in rice using markers. This gene was found to express in different organs. But the relevant GFP was located in chloroplasts. dnrl1 mutants has been reported to contain a lower level of aromatic amino acids.

The results are interesting and could be suitable for the readers of IJMS. However, the manuscript needs revision before publication.

Discussion section needs major revisions. Corroborate your results with previous findings.

Write a conclusion of your findings.

How relative expression of genes was calculated-describe in M&M. Did you test primer specificity and efficiency?

Article needs moderate grammar editing.

Tables and figures must stand alone. Define every components in figure 7 and other in figures.

Lines 40-41: meaning not clear.

Line 58: bad grammar

Formatting of references are not always consistent.

Reviewer 2 Report

Review Comments

In this manuscript the authors have isolated a rice Dynamic Narrow-Rolled leaf 1 (dnrl1) mutant showing reduced width of leaf blades, rolled leaves and lower chlorophyll content. The narrow-rolled leaf phenotype was resulted from the reduced number of small longitudinal veins per leaf smaller size and irregular arrangement of bulliform cells compared with the wild-type. Sequence analysis revealed that a single base substitutions (T-A) was detected in dnrl1, leading to a single amino acid change (L376H) in the coding protein. The mutation led to a lower level of expression of DNRL1 as well as the lower activity of DAHPS in the mutant. Genetic complementation and over-expression of DNRL1 could rescue the narrow-rolled phenotype. The lower level of chlorophyll in dnrl1 was associated with down-regulation of genes responsible for chlorophyll biosynthesis and photosynthesis.

In general, the study shows some interesting results. However, there are some important improvements needed for this manuscript as shown below;

The manuscript needs English language editing throughout the manuscript.

Introduction

The introduction needs further information covering the literature and up-to-date information highlighting the current drawback on this topic. The research question and hypothesis should be discussed in detail. The authors should also cite the recent literature.

Material and Methods

Methods should be explained in more detail, especially the methods used for genetic manipulation, overexpression, chlorophyll measurement, ..etc.

Statistical analysis should be clearly highlighted.

Results and Discussion

The discussion should be further improved and interpreted with the results to reveal the significant findings and how these results could help enhancing the knowledge on this topic.

Literature

Recent literature should be included to reveal the research significance and research question as well.

Reviewer 3 Report

  This manuscript entitled “Characterization of a novel rice dynamic narrow-rolled leaf mutant with deficient in aromatic amino acids” by Wang, et al. describes that the cloning and characterization of rice DNRL1 gene. DNRL1 encodes a putative 3-deoxy-7-phosphoheptulonate synthase (DAHPS) which is the first key enzyme of the shikimate pathway for biosynthesis of aromatic amino acids. Loss-of-function mutants of DNRL1 show narrow and rolled leaf with a slight green color. Anatomical studies revealed that the mutant plants had fewer small longitudinal veins, small bulliform cells and lower chlorophyll contents, which may account for narrow-rolled, and light green leaf phenotype in dnrl1.

 The authors showed that lower activity of DAHPS and significantly reduced levels of aromatic amino acids in the mutant These resultsindicate that DNRL1 plays a critical role in leaf morphogenesis by mediating the biosynthesis of amino acids. In addition, expression analysis indicate that DNRL1 is expressed in all organs with varied extents, and leaf morphology-associated genes in the mutants revealed that DNRL1 possibly regulates leaf morphology by a different pathway of previously reported regulators.

  I found that the manuscript is potentially interesting and including novel findings, because the study showed evidences that some phenotypes of dnrl1 could be explained by alternation of shikimate pathway. I think that the project described in the manuscript is nice and the findings presented in the manuscript should contribute to our understanding of developmental regulations underlying change of aromatic amino acids and their secondary metabolites. However, I found some aspects that need to be addressed. My comments are follows.

The authors should follow the gene nomenclature system for rice (McCouch SR. Rice (2008) 1:72–84). Both dnrl1and DNRL1 as a mutant and gene name are not appropriate. In addition, I cannot fully understand why the mutant was designated  as ‘dynamic’ narrow-rolled leaf. What traits of the mutant is dynamic?

The authors showed that the level of the activity of DAHPS in the mutant was reduced to the half of the wild type. Is the reduction due to the reduced activity or expression level of OsDAHP1? Alternatively, the catalytic activity of OsDAHP1 was completely abolished, but does OsDAHP2 compensate the rest of DAHPS activity?

Authors should clarify;

whether and how much the amino acid substitution of the mutant affects the DAHPS activity. whether OsDAHP2 product has enough activity of DAHPS and expression level of OsDAHP2 in the various tissue.

In addition, the author should discuss why the expression level of OsDAHP1 was reduced in the mutant.

The authors described in the text that IAA treatment could not restore the mutant phenotype. The results indicates that the phenotypes of the mutants might not be related to the auxin but other compounds derived from aromatic amino acids. I think that it is not bad idea to examine whether treatment of each aromatic amino acid on the mutant can rescue the phenotype or not.

The authors described in the text that OsDnrl1-GFP is perfectly co-localized with chloroplast. However, I do not think that the data indicates the perfect co-localized signal, because green and red signal are seen in the merged panels of the Fig6.

Finally,  the section of discussion is not refined. There are repetitive descriptions of the introduction are seen in the first half and second half paragraph.

Round 2

Reviewer 1 Report

Please insert references for calculating relative expression of gene and statistical analysis.

Improve discussion further based on results.

Author Response

Please insert references for calculating relative expression of gene and statistical analysis.

Response:Thanks. We have included the reference for calculating the relative expression of the genes tested.

Improve discussion further based on results.

Response: We are grateful to the comment and suggestion, and fully agree that the discussion can be further improved while we think the main issues have been addressed in certain degrees and hopefully it is allowed in this rvised version of manuscript.

Reviewer 2 Report

The authors have addressed my concerns and the manuscript has been greatly improved 

Author Response

The authors have addressed my concerns and the manuscript has been greatly improved 

Response: We are grateful to the comment!

Reviewer 3 Report

This version of manuscript was improved and I have no specific comment on my previous suggestions.

Author Response

This version of manuscript was improved and I have no specific comment on my previous suggestions.

Response: We are grateful to your comment !